# Diagnosing Intramammary Infection: Meta-Analysis and Mapping Review on Frequency and Udder Health Relevance of Microorganism Species Isolated from Bovine Milk Samples

**DOI:** 10.3390/ani12233288

**Published:** 2022-11-25

**Authors:** Daryna Kurban, Jean-Philippe Roy, Fidèle Kabera, Annie Fréchette, Maryse Michèle Um, Ahmad Albaaj, Sam Rowe, Sandra Godden, Pamela R. F. Adkins, John R. Middleton, Marie-Lou Gauthier, Greg P. Keefe, Trevor J. DeVries, David F. Kelton, Paolo Moroni, Marcos Veiga dos Santos, Herman W. Barkema, Simon Dufour

**Affiliations:** 1Faculté de Médecine Vétérinaire, Université de Montréal, Saint-Hyacinthe, QC J2S 2M2, Canada; 2Mastitis Network, Saint-Hyacinthe, QC J2S 2M2, Canada; 3Research Group Op+Lait, Saint-Hyacinthe, QC J2S 2M2, Canada; 4Faculty of Science, The University of Sydney, Camden, NSW 2570, Australia; 5Department of Veterinary Population Medicine, University of Minnesota, St. Paul, MN 55108, USA; 6Department of Veterinary Medicine and Surgery, University of Missouri, Columbia, MO 65211, USA; 7Laboratoire de Santé Animale, Ministère de l’Agriculture, des Pêcheries et de l’Alimentation du Québec (MAPAQ), Saint-Hyacinthe, QC J2S 2M2, Canada; 8Department of Health Management, Atlantic Veterinary College, University of Prince Edward Island, Charlottetown, PE C1A 4P3, Canada; 9Department of Animal Biosciences, University of Guelph, Guelph, ON N1G 2W1, Canada; 10Ontario Veterinary College, University of Guelph, Guelph, ON N1G 2W1, Canada; 11Animal Health Diagnostic Center, Quality Milk Production Services (QMPS), Cornell University, Ithaca, NY 14853, USA; 12Dipartimento Medicina Veterinaria e Scienze Animali, Universita’ Degli Studi di Milano, 26900 Lodi, Italy; 13Department of Animal Nutrition and Production, School of veterinary Medicine and Animal Sciences, University of São Paulo, Pirassununga 13630-000, SP, Brazil; 14Faculty of Veterinary Medicine, University of Calgary, Calgary, AB T2N 4N1, Canada

**Keywords:** cattle, milk microbiology, MALDI-TOF, mastitis, species-specific prevalence

## Abstract

**Simple Summary:**

Innovations in veterinary medicine diagnostic methods can help to better identify the microorganisms causing bovine mastitis. Matrix-assisted laser desorption/ionization time-of-flight (MALDI-TOF) mass spectrometry is such an innovation. This tool can identify many different microorganisms cultured from bovine milk samples to species-level. However, not all these microorganisms are necessarily pathogenic to the cow udder. Using 93,353 milk samples from different countries, we aimed to determine the diversity and proportion of different microorganisms cultured from bovine milk of apparently healthy cow mammary glands and of clinical mastitis cases, and identified by MALDI-TOF. Moreover, we highlighted the current knowledge gaps on the effect of these microorganisms on udder health. We revealed a great diversity of microorganisms in both types of samples, but, for most species (206 of 294), little literature regarding their udder health relevance was available. However, some microorganisms with little or no scientific literature were frequently isolated from clinical mastitis samples; thus, suggesting that they may be relevant in terms of udder health. For other species, more research is needed to clarify their role.

**Abstract:**

Matrix-assisted laser desorption/ionization time-of-flight (MALDI-TOF) mass spectrometry provides accurate species-level identification of many, microorganisms retrieved from bovine milk samples. However, not all those microorganisms are pathogenic. Our study aimed to: (1) determine the species-specific prevalence of microorganisms identified in bovine milk of apparently healthy lactating quarters vs. quarters with clinical mastitis (CM); and (2) map current information and knowledge gaps on udder health relevance of microorganisms retrieved from bovine milk samples. A mixed study design (meta-analysis and mapping review) was chosen. We gathered several large Canadian, US and Brazilian data sets of MALDI-TOF results for organisms cultured from quarter milk samples. For meta-analysis, two datasets (apparently healthy quarters vs. CM samples) were organized. A series of meta-analyses was conducted to determine microorganisms’ prevalence. Then, each species reported was searched through PubMed to investigate whether inflammation (increased somatic cell count (SCC) or signs of CM) was associated with microorganism’s recovery from milk. A total of 294 different species of microorganisms recovered from milk samples were identified. Among 50,429 quarter-milk samples from apparently healthy quarters, the 5 most frequent species were *Staphylococcus chromogenes* (6.7%, 95% CI 4.5–9.2%), *Aerococcus viridans* (1.6%, 95% CI 0.4–3.5%), *Staphylococcus aureus* (1.5%, 95% CI 0.5–2.8%), *Staphylococcus haemolyticus* (0.9%, 95% CI 0.4–1.5%), and *Staphylococcus epidermidis* (0.7%, 95% CI 0.2–1.6%). Among the 43,924 quarter-milk CM samples, the 5 most frequent species were *Escherichia coli* (11%, 95% CI 8.1–14.3%), *Streptococcus uberis* (8.5%, 95% CI 5.3–12.2%), *Streptococcus dysgalactiae* (7.8%, 95% CI 4.9–11.5%), *Staphylococcus aureus* (7.8%, 95% CI 4.4–11.9%), and *Klebsiella pneumoniae* (5.6%, 95% CI 3.4–8.2%). When conducting the PubMed literature search, there were 206 species identified by MALDI-TOF for which we were not able to find any information regarding their association with CM or SCC. Some of them, however, were frequently isolated in our multi-country dataset from the milk of quarters with CM (e.g., *Citrobacter koseri*, *Enterococcus saccharolyticus*, *Streptococcus gallolyticus*). Our study provides guidance to veterinarians for interpretation of milk bacteriology results obtained using MALDI-TOF and identifies knowledge gaps for future research.

## 1. Introduction

Mastitis is a major welfare and economic concern on dairy farms [1]. Mastitis is classified as subclinical (SCM) or clinical (CM) [2]. Intramammary infections (IMI) are the main cause of mastitis in dairy cows worldwide.

Clinical mastitis is identified by observation of visible clinical signs (pain, redness, heat, and swelling of the mammary gland tissues, and/or visual abnormalities of the milk from the affected mammary gland quarter, and/or systemic signs of illness, such as hyperthermia, loss of appetite, etc.), while the identification of SCM requires additional diagnostic tests. Milk somatic cell count (SCC) is used commonly to detect SCM. Somatic cells consist of epithelial cells and immune cells. The milk SCC of healthy mammary gland quarters is generally less than 100,000 cells or even 50,000 cells per mL of milk [3,4]. Elevation of the SCC is an indicator of inflammation, caused by the increase in the fractions of immune cells, mainly neutrophils. It thus indicate an immune response due to the presence of an IMI [2]. There are direct and indirect SCC measurement methods [5]. Among the direct methods, flow cytometry, direct microscopic count, or scanning electrochemical microscopy count can be used [6,7,8]. The most convenient and commonly used indirect method is probably the California mastitis test, which is based on the visual evaluation of the milk viscosity following addition of a reactive [9,10,11]. Otherwise, measuring milk electrical conductivity [6,12,13], or the identification of protein biomarkers (e.g., interleukins, lactoferrin, cytokines, amyloid A, haptoglobin) using enzyme-linked immunosorbent assay method could help to identify the presence of inflammation [14]. However, all these methods are not useful to identify the causative pathogens that cause the CM or SCM.

Pathogens causing mastitis can be identified in milk using conventional microbiological culture (different selective media and biochemical methods), using DNA-based molecular (e.g., PCR, 16S rRNA sequencing, WGS), proteomic (e.g., matrix-assisted laser desorption/ionization time-of-flight mass spectrometry; MALDI-TOF), or other methods [15,16,17,18,19]. Recently, MALDI-TOF method was introduced in many diagnostic laboratories [5]. This proteomics analysis tool is accurate, fast, and cost effective. It allows precise identification to the species-level for many microorganisms present in milk [20]. Thus, compared to conventional biochemical tests, using MALDI-TOF method improved the ability to identify the microorganisms isolated in culture, to species-level, with accuracy (i.e., an increased typeability) [21,22]. We are now able to better identify a very large number of microorganisms, mainly bacteria, fungi, and algae isolated from milk samples of apparently healthy milking quarters and from quarters with CM.

Some microorganisms have been clearly associated in the literature with SCM and/or CM, and, when they are retrieved from a milk sample, most will conclude that an IMI is present. For instance, Current Concepts of Bovine Mastitis [2] listed the following microorganisms as important mammary gland pathogens: *Staphylococcus aureus*, *Streptococcus agalactiae*, *Streptococcus uberis*, *Streptococcus dysgalactiae* and other *Streptococcus*-like microorganisms, *Mycoplasma bovis* and other *Mycoplasma* spp., *Corynebacterium bovis*, *Escherichia coli*, *Klebsiella* spp., *Enterobacter* spp., *Citrobacter* spp., *Enterococcus faecalis*, *Enterococcus faecium*, non-aureus staphylococci (NAS) (*Staphylococcus chromogenes*, *Staphylococcus hyicus, Staphylococcus warneri, Staphylococcus epidermidis, Staphylococcus cohnii, Staphylococcus simulans, Staphylococcus xylosus, Staphylococcus sciuri, Staphylococcus saprophyticus*), *Pseudomonas aeruginosa*, *Trueperella pyogenes*, *Nocardia* spp., mycobacteria, *Serratia* spp., *Bacillus cereus*, yeasts (*Candida*), molds, and algae (*Prototheca*).

However, for many of the microorganisms’ species that can be retrieved from milk samples, the literature regarding their association with IMI and their effect on mammary gland health is still scarce. In veterinary medicine, the term IMI is often misused to describe the simple presence of microorganisms in milk. However, according to the Merriam-Webster Medical dictionary [23], the term infection rather describes “*The invasion and multiplication of microorganisms such as bacteria, viruses, and parasites that are not normally present within the body. An infection may cause no symptoms and be subclinical, or it may cause symptoms and be clinically apparent. […] Microorganisms that live naturally in the body are not considered infections. …*”. Therefore, when bacterial, algae, fungal cells, or DNA are retrieved from a milk sample, we cannot necessarily conclude that this constitutes an IMI. Nevertheless, in many instances, retrieval of a microorganism not normally present in milk (or usually present, but in very small concentrations) will coincide with measurable or visible signs of inflammation of the udder (mastitis). Such cases would match the proposed Merriam-Webster infection definition and could be considered IMI. Thus, the microorganism’s species that can nowadays be identified from milk samples using MALDI-TOF should be considered relevant, from an udder-health perspective, if and only if they can be associated with a measurable inflammation (either an increase in SCC or visible signs of CM).

To support the interpretation of milk microbiological analyses conducted using MALDI-TOF technology, guidelines are needed regarding the udder health relevance of the different species that are commonly reported using this method. The objectives of our study were, therefore, to: (1) determine the species-specific prevalence of microorganisms identified in bovine milk of apparently healthy lactating quarters vs. quarters with CM using MALDI-TOF; and (2) map current information and knowledge gaps on udder health relevance of microorganisms retrieved from bovine milk samples.

## 2. Materials and Methods

For the current study, we used a hybrid study design consisting of series of individual patient data meta-analyses and of mapping reviews. Our protocol was developed and described using guidelines from the Preferred Reporting Items for Systematic Reviews and Meta-Analyses. More specifically, since there is no specific guidelines for mapping review, we adapted those provided in the extension for Scoping Reviews [24]. The protocol was made available on an open access web depository (SYREAF; Systematic reviews for animals & food; http://www.syreaf.org/contact/, accessed on 15 July 2022) prior to conducting the study.

### 2.1. Study Design and Data Collection

Firstly, several authors that had previously published research studies performing identification of microorganisms detected in bovine milk samples using MALDI-TOF (mainly located in North and South America, Europe, New-Zealand, and Australia) were contacted. An invitation was also sent through a larger mailing list used by international mastitis scientists (n = 126), the Mastitis Research Workers mailing list. Potential collaborators were asked to provide any research or diagnostic original databases where MALDI-TOF was used for identification of microorganisms and where either: (1) quarter-milk samples were collected on apparently healthy lactating quarters (without any pre-selection criteria for quarters sampled: e.g., not only from high SCC quarters, or only from quarters selected because of a known IMI history); or (2) quarter-milk samples were collected from CM cases. The datasets had to be organized in an individual cow mammary quarter data format (i.e., an individual patient format) with indication whether the sample was from an apparently healthy milking cow mammary quarter or from a quarter with CM, along with the bacteriological result, including the MALDI-TOF identification. Thus, milk samples yielding no growth were also included. Whenever possible, the number of samples that were deemed to be contaminated were also reported, although these were later considered as uninformative and excluded from subsequent analyses. General information (e.g., geographical region, number of herds) regarding the datasets were also obtained from the dataset’s owner and/or from previously published articles using these research data.

### 2.2. Meta-Analysis

The different databases obtained were organized in two new datasets: one including data from all studies on apparently healthy milking quarters and one including data from the CM studies. In these datasets, we summarized, for each study and each result observed, the total number of samples where the microorganism (or no growth) was observed, and the total number of samples collected.

To determine the species-specific prevalence of microorganisms identified in bovine milk of apparently healthy lactating quarters and from quarters with CM, random effects meta-analyses were used. First, for each dataset separately (mammary quarters with CM vs. apparently healthy lactating quarters), and for each microorganism (identified by species, or by genus, whenever MALDI-TOF could not identify to species-level) and for no growth results, a meta-analysis was used for estimating a summary measure of the proportion of the samples with this bacteriology result.

Moreover, we hypothesized that, for some microorganisms, the prevalence may vary as a function of the country of origin. This latter covariate was, thus, investigated using meta-regression. Statistical significance was determined at a *p*-value < 0.05. Heterogeneity between study was assessed by the I^2^ statistic. The series of meta-analyses and meta-regressions were performed using the metafor package [25] of R (version 4.0.5; R Foundation for Statistical Computing, Vienna, Austria). The inverse variance approach was used to assign weights to the studies, and the prevalence was estimated using a logit distribution. The Hartung-Knapp adjustment for random effects model with Freeman-Tukey double arcsine transformation were used to assure the variance stabilization after logit transformation.

### 2.3. Mapping Review

As a second step, we tried to identify the microorganisms for which there was no or very little peer-reviewed literature on their association with udder health (i.e., association with an increased SCC or with CM). For this objective, a mapping review methodology was chosen. This type of review is usually conducted not only to identify key characteristics related to a specific topic, but, most importantly, to identify and analyze knowledge gaps [26]. The research questions were formulated as: (1) what is the current knowledge about the udder health pathogenicity of the microbial species found in milk samples from apparently healthy lactating mammary quarters and from milk samples from bovine CM cases?; and (2) what are the microbial species for which knowledge regarding udder health relevance is absent or scarce?

The first step of the review was to develop a comprehensive list of the microbial species (bacteria, algae, and fungi) retrieved in cow’s milk from apparently healthy lactating quarters and from quarters with CM, and identified using MALDI-TOF. To achieve this, we compiled the complete list of microorganisms identified in the multi-country datasets. Microorganisms reported in these datasets, but only identified at genus-level (e.g., *Streptococcus* spp.) or reported as a group of microorganisms (e.g., other Gram-positive) were not considered for the subsequent search of the literature. Moreover, all species of microorganisms already described as important mastitis-causing pathogens in Current Concepts of Bovine Mastitis [2] were excluded from the subsequent literature search as this reference synthesizes the current knowledge on udder health of dairy cows. The microorganisms mentioned in this document are already well-known mastitis-causing pathogens. Species listed in this reference document are: *Staphylococcus aureus*, *Streptococcus agalactiae*, *Streptococcus uberis*, *Streptococcus dysgalactiae*, *Mycoplasma bovis*, *Corynebacterium bovis*, *Escherichia coli*, *Enterococcus faecalis*, *Enterococcus faecium*, *Staphylococcus chromogenes*, *Staphylococcus hyicus*, *Staphylococcus warneri*, *Staphylococcus epidermidis*, *Staphylococcus cohnii*, *Staphylococcus simulans*, *Staphylococcus xylosus*, *Staphylococcus sciuri*, *Staphylococcus saprophyticus*, *Pseudomonas aeruginosa*, *Trueperella pyogenes* and *Bacillus cereus*. We thus made an a priori assumption that a lot of scientific literature confirming their udder health relevance was already available and that no major knowledge gap would be identified for these specific microorganisms.

A general search strategy was developed to identify peer-reviewed manuscripts reporting on a given microbial species (the exposure) and a measure of inflammation (the outcome) in dairy cows (the population). The search terms used were genus and species of a microorganism (for the exposure), somatic cell* OR intramammary infection OR mastitis (for the outcome), and cattle OR cow OR *bos taurus* OR bovin* (for the population). For instance, the search strategy for *Lactococcus lactis* was:

(*Lactococcus lactis*) AND ((somatic cell*) OR (intramammary infection) OR (mastitis)) AND ((cattle OR cow OR (bos taurus) OR bovin*))

Since the goal of this mapping review was to quickly map the relative amount of knowledge available on each microbial species, and given the large number of species to review, only one bibliographic database, Medline (PubMed), was searched. To find potentially relevant papers, the different Medline searches (one per microorganism species) were initially conducted sequentially over a 1-year time period (between 21 July 2020 and 20 June 2021). Each microbial species was searched independently, by two reviewers, in parallel. When this initial search was completed, the bibliographic search was updated for all species on the same day (16 July 2021) to add any new articles. Documents from 1976 and after, written in English, French, or German were considered for the review. Again, because the goal was to identify knowledge gaps (vs. reporting all the literature on a microbial species), the search strategy was refined whenever the search strategy for a given microbial species yielded more than 30 articles to review. As a first attempt to refine the search strategy, we ensured that the microbial species were mentioned in the title or the abstract (vs. the text) of the articles. The search strategy was thus modified as follows (using the previous example):

*Lactococcus lactis* [Title/Abstract] AND ((somatic cell*) OR (intramammary infection) OR (mastitis)) AND (cattle OR cow OR (bos taurus) OR bovin*)

Whenever this more specific search strategy still generated more than 30 articles, we specifically looked for review, meta-analysis, or systematic review types of articles. When such article types were available, and when association with SCC or CM were reported, then only these were reviewed. Whenever the review-type articles did not report on association with SCC or CM, all articles from the previously described title and abstract search strategy were reviewed. Finally, if the articles from the title and abstract search strategy did not report associations with SCC or CM, then all articles listed by the first, broader search strategy were reviewed. This approach was used to ensured that we would not wrongly conclude that literature on association between a given bacterial species and udder health was absent or scarce.

The abstracts and full texts of the listed articles were assessed independently by teams of two reviewers (DK, SD, JPR, MMU, AA, AF, FK) to make sure that the study was conducted using milk samples obtained from dairy cows and to extract the data regarding associations with SCC or CM. For each bacterial species, a list of the articles presenting SCC or CM results was constituted. All data reporting on SCC (SCC, somatic cell score, CMT, or other measurements) of the affected quarter or cow, or on SCC difference between infected and healthy quarters or cows, or on presence of clinical signs of mastitis were extracted and summarized by article. Regarding presence of clinical signs, when the authors of the reviewed article solely used wording such as samples from “mastitic cows” or from “mastitis”, without explicitly referring to clinical signs of disease or to “clinical” mastitis, then we did not consider that these were necessarily CM. We did so, because these terms are also often used for intramammary infections occurring with or without quantifiable signs of inflammation or SCC information.

## 3. Results

In total, 126 researchers were contacted by e-mail. Eight of the invited researchers were able to provide one or multiple datasets with MALDI-TOF results that fit the inclusion criteria. All these MALDI-TOF analyses were conducted using the equipment MALDI Biotyper^®^ system (by Bruker Daltonics Inc., Billerica, MA, USA).

### 3.1. Species-Specific Prevalence of Microorganisms in Bovine Milk

We obtained 10 datasets with MALDI-TOF results from apparently healthy lactating mammary quarters milk samples, and 8 datasets from quarters with CM. Detailed descriptions of the apparently healthy quarters and CM quarters datasets are provided in Table 1 and Table 2, respectively. Briefly, all datasets were assembled using data from North (Canada and US) and South America (Brazil) regions. Our inferences on species-specific prevalence should, therefore, be restricted to these regions.

A total of 53,684 milk sample results from 113 herds were available for studying prevalence of microorganisms in milk from apparently healthy milking quarters. Of these, 50,429 samples (93.9%) could be included after exclusion of contaminated samples. Among the included samples, no colony forming units (CFU) could be observed after incubation on 34,415 samples (68.2%). All samples originated from recently conducted research projects (2017 to 2021).

For milk samples obtained from CM cases, 45,272 milk sample results from 697 herds were available for studying prevalence of microorganisms. Of these, 43,924 samples (97.0%) could be included after exclusion of contaminated samples. Among the included samples, no CFU could be observed after incubation on 17,401 samples (39.6%). Six of the CM datasets obtained originated from research projects and two were obtained from diagnostic laboratories (Quality Milk Production Services (QMPS), Ithaca, NY, US; Centre de diagnostic vétérinaire de l’Université de Montréal, CDVUM, St-Hyacinthe, QC, Canada).

A total of 294 different species of microorganisms recovered from milk samples were identified among the two assembled datasets (227 species in the apparently healthy milking quarters database and 178 species in the CM database). Prevalence of the different microorganisms identified using MALDI-TOF in bovine milk of apparently healthy lactating quarters and estimated using random meta-analyses are reported in Table 3 for microorganisms with an estimated prevalence ≥0.1%. Among the quarter-milk samples from apparently healthy quarters, only 26 groups of microorganisms had an estimated prevalence ≥0.1% with 11 of them being related to the *Staphylococcus* genus, though one of the microorganisms, *Staphylococcus sciuri*, was recently reassigned to the novel genus *Mammaliicoccus* with *Mammaliicoccus sciuri* as the type species [36]. The five most prevalent species were *Staphylococcus chromogenes* (6.7%, 95% CI 4.5–9.2%), *Aerococcus viridans* (1.6%, 95% CI 0.4–3.5%), *Staphylococcus aureus* (1.5%, 95% CI 0.5–2.8%), *Staphylococcus haemolyticus* (0.9%, 95% CI 0.4–1.5%), and *Staphylococcus epidermidis* (0.7%, 95% CI 0.2–1.6%). For most species, the estimated prevalence did not differ between countries.

Meta-analysis-derived prevalence estimates of microorganisms retrieved from CM samples are presented in Table 4. Among the 43,924 quarter-milk CM samples, 43 groups of microorganisms had an estimated prevalence ≥0.1%. The most frequent genus was, again, the *Staphylococcus* genus (n = 11, when including *Staphylococcus sciuri*), followed by *Streptococcus* genus (n = 7). The 5 most frequent species in CM samples were *Escherichia coli* (11%, 95% CI 8.1–14.3%), *Streptococcus uberis* (8.5%, 95% CI 5.3–12.2%), *Streptococcus dysgalactiae* (7.8%, 95% CI 4.9–11.5%), *Staphylococcus aureus* (7.8%, 95% CI 4.4–11.9%), and *Klebsiella pneumoniae* (5.6%, 95% CI 3.4–8.2%). Again, for most microorganisms, the estimated prevalence was not significantly affected by the country of origin.

For microorganisms retrieved from both apparently healthy lactating and CM samples with prevalence ≥0.1%, the I^2^ statistic was always greater than 50%, indicating substantial heterogeneity among datasets for these microorganisms.

All other microorganisms reported, but for which an estimated prevalence <0.1% was obtained are listed in Appendix A (https://doi.org/10.5683/SP3/VOUUYO, accessed on 15 July 2022). The complete meta-analyses results (forest plots, funnel plots, influential plots, leave out plots, and meta-regression by country) are also available as Appendix A (https://doi.org/10.5683/SP3/VOUUYO).

### 3.2. Mapping Review

We searched for available literature for a total of 273 microorganism species (after excluding the 21 species already described as mastitis-causing pathogens in Current concepts of bovine mastitis [2]). The scientific literature retrieved and assessed are summarized in Table 5 for species where an association with visible CM symptoms (n = 13), high SCC (n = 12), or both (n = 39) was reported in the literature. Moreover, the journal articles reporting these associations are listed as Appendix A (https://doi.org/10.5683/SP3/VOUUYO). Among these 64 species, 14 (21.8%) and 8 (12.5%) were isolated from ≥0.1% of samples of quarters with CM, or from samples of apparently healthy milking quarters, respectively. There were 47 (73.4%) species for which we could detect an association with high SCC and/or CM symptoms in the literature reviewed, but for which the estimated prevalence in our study was <0.1% (Table 5 and Appendix A (https://doi.org/10.5683/SP3/VOUUYO)).

For most species (n = 206; 70.1%) isolated from milk samples, we were not able to find any literature reporting association with CM, nor SCC using Medline (PubMed) research database tool. Some of these species were, however, frequently identified in CM samples in our multi-country CM dataset: e.g., *Citrobacter koseri*, *Enterococcus saccharolyticus*, *Streptococcus gallolyticus*. *Corynebacterium amycolatum.* Thus, an important knowledge gap was identified for most species, but more specifically for these latter species.

## 4. Discussion

For many microorganisms isolated from bovine milk and identified using MALDI-TOF, their udder health relevance is not clear. For instance, some of these microorganisms could be commensal microorganisms, thus, being present in milk, but without harming the mammary gland. In other cases, these microorganisms could have colonized the teat canal, teat end, or even the mammary gland cistern, but, again, without causing damage to the mammary epithelial cells, nor eliciting a measurable immune response [40,41]. Finally, we cannot exclude that, in some case, finding these microorganisms in milk could result from a contamination of the milk sample at sampling. Nevertheless, when including the 21 species already reported as important mastitis-causing pathogens by Current concepts of bovine mastitis [2], our results would confirm that at least 85 different microorganisms could trigger an elevation of SCC and/or clinical signs of mastitis in dairy cattle.

On the other hand, only 32 of these species (15 of the 21 from Current concepts of bovine mastitis [2] and 17 from our study) would have an estimated prevalence ≥0.1% based on our multi-country dataset. Within the ones listed as important pathogens in Current concepts of bovine mastitis [2], *Streptococcus agalactiae, Mycoplasma bovis, Staphylococcus hyicus, Staphylococcus warneri, Staphylococcus cohnii*, and *Bacillus cereus* all had an estimated prevalence <0.1%. More particularly, *Staphylococcus agalactiae*, which used to be a common mastitis pathogen was found in only 31 milk samples. Similarly, *Mycoplasma bovis* was never retrieved, but this could be due to the low sensitivity of conventional milk culture for this microorganism, as no specific routine mycoplasma culture medium was used in any study included in project [42,43]. Species not already listed in Current concepts of bovine mastitis [2], for which we were able to find in the literature associations with SCC and/or CM, and for which a prevalence ≥0.1% was estimated were: *Aerococcus viridans*, *Bacillus licheniformis*, *Bacillus pumilus*, *Candida krusei*, *Candida rugosa*, *Enterobacter cloacae*, *Klebsiella oxytoca*, *Klebsiella pneumoniae* (note that *Klebsiella* spp. are mentioned in this reference manual, but specific species are not given)*, Lactococcus garvieae*, *Lactococcus lactis*, *Pasteurella multocida*, *Staphylococcus auricularis*, *Staphylococcus equorum*, *Staphylococcus haemolyticus*, *Staphylococcus hominis*, *Streptococcus equinus*, and *Streptococcus lutetiensis* (note that *Streptococcus*-like species are mentioned in this reference manual, but specific species are not given).

In our study, all milk samples for the database on apparently healthy lactating mammary quarters originated from scientific research (vs. regular diagnostic activities). Thus, we had very precise information on the cow and quarter selection process used for the data collection. To assemble the database for the meta-analysis, we only chose situations where there was no preselection of quarters or cows for previous history of IMI or high SCC. This was done to ensure that we could provide prevalence estimates for a population of apparently healthy lactating quarters. Nevertheless, the sampling method could differ among the 8 studies. This effect could explain, for instance, the difference in contaminated samples proportion, as well as in proportion of samples with no growth. Furthermore, differences in bacteriological culture techniques and agar lectures prior to MALDI-TOF identification could possibly create differences between studies (e.g., different milk sample storage methods before culturing and the use of different culture media) [44,45,46].

While 6 of 10 datasets from CM cases originated from research projects, most of the individual patient data were obtained from regular diagnostic activities (41,651 of 43,924 samples included). For these latter results, the sampling method could not be controlled. This could explain the difference in prevalence of contaminated samples among the datasets.

Within CM samples, a proportion of the samples classified as negative (where no growth was observed) could possibly be explained by the limited shedding of the pathogen. For instance, during CM caused by *E. coli,* bacterial cells may be eliminated from the udder tissue, but the lipopolysaccharide endotoxin will still have a harming effect on the mammary gland and trigger clinical signs, thus leading to sampling when viable bacterial cells are not present anymore [47]. Furthermore, the protocol for bacteriological culture of the milk samples from CM cases could vary from one laboratory to another: the use of different culture media or the use of a supplementary enriched culture when no growth was observed after 24 h incubation of the CM samples may have impacted culture yield and the results interpretation [35,48].

All results were obtained by the same mass spectrometry method and using similar equipment (MALDI Biotyper^®^ system by Bruker Daltonics Inc., Billerica, MA, USA). However, between 2017 and 2021 (the period where these samples were analyzed), the library of reference spectra was frequently updated by the manufacturer from the MBT-BDAL-6903 (containing 6903 mass spectra) to the MBT-BDAL-10833 (containing 10,833 mass spectra for 3893 microorganism species). It is, therefore, possible that some microorganisms could not be identified to species-level in older studies compared to the most recent ones.

Furthermore, for the Canadian datasets (Kurban et al., personal communication, Kabera et al., 2020 [27], Fréchette et al., 2021 [34], and CDVUM) and for some US datasets (Wattenburger et al., 2020 [28], Middleton (personal communication), Ankney et al., 2020 [32]) additional custom reference spectra libraries that could possibly improve identification of some NAS were used [49]. This could possibly have created difference between countries in the estimated prevalence of some staphylococcal species. When the Canadian custom library was validated, however, the MBT-BDAL-5627 library was used, and this latter library could not identify to the species-level a number of staphylococcal isolates. Therefore, Cameron et al. [49] added a number of reference mass spectra for: *Staphylococcus agnetis*, *Staphylococcus cohnii*, *Staphylococcus devriesei*, *Staphylococcus* (now *Mammaliicoccus*) *fleurettii*, *Staphylococcus gallinarum*, *Staphylococcus warneri*, and *Staphylococcus xylosus* (at the time, none of these were included in the MBT-BDAL-5627), and also some extra *Staphylococcus chromogenes* and *Staphylococcus haemolyticus* (these last two were included in the MBT-BDAL-5627, but in limited numbers) [49]. However, many mass spectra were also added by the manufacturer in their main library. For instance, the more recent MBT-BDAL-10833, contains a number of reference mass spectra for *Staphylococcus cohnii* (n = 9), *Staphylococcus devriesei* (n = 1), *Staphylococcus* (now *Mammaliicoccus*) *fleurettii* (n = 1), *Staphylococcus gallinarum* (n = 5), *Staphylococcus warneri* (n = 8), *Staphylococcus xylosus* (n = 6), *Staphylococcus chromogenes* (n = 6), and *Staphylococcus haemolyticus* (n = 10). Therefore, it is not clear anymore whether using the custom library [49] would alter at all the diagnostic ability compared to solely using the manufacturer’s library. Nevertheless, although its prevalence was very low, *Staphylococcus agnetis* was only identified in the studies where custom libraries including a mass spectra profiles for this bacterial species were used, thus suggesting that the libraries used may have played a role.

Finally, bovine milk microbiota and most common pathogenic agents certainly vary between regions, or even within a country [50]. The data available in our study were strictly from studies and laboratories in Canada, US and Brazil. Therefore, the results of our research are not necessarily generalizable to dairy farms from Europe, Asia, Africa, or Oceania.

## 5. Conclusions

Our results indicate that: (1) a large number of microbial species can be isolated from bovine milk samples and identified using MALDI-TOF; (2) species distribution varies by sample type (apparently healthy quarter vs. CM quarter); (3) for many of these species, there is little or no literature regarding their potential for eliciting an inflammatory response in the udder; and (4) for some species frequently isolated from CM there is no literature available regarding their pathogenicity.

Our study provides guidance to help veterinarians, dairy producers and dairy advisors in interpreting milk bacteriological analyses conducted using MALDI-TOF and has highlighted several knowledge gaps that should be addressed in future research.

## Figures and Tables

**Table 1 animals-12-03288-t001:** General characteristics of the datasets on microorganisms identified using MALDI-TOF from apparently healthy bovine lactating quarters.

Study	Country	No. Herds	No. Milk Samples	No. Contaminated Samples (% of Total Samples)	No. Included Samples ^a^	No. Samples with No Growth (% of Included Samples)
Kurban et al. (personal communication)	Canada	5	16,462	935 (5.7)	15,527	8935 (57.5)
Kabera et al., 2020 [27]	Canada	9	6215	140 (2.3)	6,075	4593 (75.6)
Wattenburger et al., 2020 [28]	US	1	653	259 (39.7)	394	188 (47.7)
Rowe et al., 2019 [29]	US	78	11,560	1112 (9.6)	10,448	8249 (79.0)
Rowe et al., 2020 [30]	US	7	10,200	462 (4.5)	9738	6376 (65.5)
Rowe et al., 2020 [31]	US	5	6525	165 (2.5)	6360	4796 (75.4)
Middleton (personal communication)	US	1	233	6 (2.6)	227	189 (83.3)
Ankney et al., 2020 [32]	US	1	147	11 (7.5)	136	84 (61.8)
Ankney et al., 2020 [32]	US	1	629	55 (8.7)	574	380 (66.2)
Garcia et al., 2021 [33]	Brazil	5	1060	110 (10.4)	950	625 (65.8)
**Total**		113	53,684	3255 (6.1)	50,429	34,415 (68.2)

^a^ Number of samples after exclusion of contaminated (uninformative) samples.

**Table 2 animals-12-03288-t002:** General characteristics of the datasets on microorganisms identified using MALDI-TOF from dairy cow milk samples from quarters with clinical mastitis.

Study	Country	No. Herds	No. Milk Samples	No. Contaminated Samples (% of Total Samples)	No. Included Samples ^a^	No. Samples with No Growth (% of Included Samples)
Kurban et al. (personal communication)	Canada	5	348	19 (5.5)	329	45 (13.7)
Kabera et al., 2020 [27]	Canada	9	68	1 (1.5)	67	38 (56.7)
Fréchette et al., 2021 [34]	Canada	86	1012	132 (13.0)	880	149 (16.9)
Moroni et al. QMPS ^b^	US	36	41,301	970 (2.4)	40,331	16,551 (41.0)
CDVUM ^c^	Canada	521 ^d^	1504	184 (12.2)	1320	175 (13.3)
Veiga Dos Santos (personal communication)	Brazil	14	152	2 (1.3)	150	68 (45.3)
Granja et al., 2021 [35]	Brazil	15	448	19 (4.2)	429	192 (44.8)
Granja et al., 2021 [35]	Brazil	11	439	21 (4.8)	418	183 (43.8)
**Total**		697	45,272	1348 (3.0)	43,924	17,401 (39.6)

^a^ Number of samples after exclusion of contaminated (uninformative) samples. ^b^ Quality Milk Production Services (QMPS), Cornell University, Ithaca, NY, USA. ^c^ Centre de diagnostic vétérinaire de l’Université de Montréal, St-Hyacinthe, QC, Canada. ^d^ Approximate number of herds.

**Table 3 animals-12-03288-t003:** Prevalence of the different microorganisms isolated from the milk of apparently healthy mammary quarters of dairy cows and identified using MALDI-TOF healthy estimated using random meta-analyses conducted on the raw data from 50,429 milk samples using 10 datasets (Canada, US and Brazil). Only microorganisms with estimated prevalence ≥0.1% are reported.

Microorganism	Prevalence (95% CI)	95% Prediction Interval ^a^
*Staphylococcus chromogenes*	6.7 (4.5–9.2)	1.1–16.4
*Aerococcus viridans*	1.6 (0.4–3.5)	0.0–10.5
*Staphylococcus aureus*	1.5 (0.5–2.8)	0.0–6.9
Unspeciated ^b^ *Corynebacterium*	1.0 (0.2–2.4)	0.0–7.4
*Staphylococcus haemolyticus*	0.9 (0.4–1.5)	0.0–3.6
Unspeciated ^b^ *Bacillus*	0.9 (0.2–2.1)	0.0–6.2
*Staphylococcus epidermidis*	0.7 (0.2–1.6)	0.0–4.7
*Corynebacterium amycolatum*	0.6 (0.0–2.0)	0.0–8.1
*Staphylococcus sciuri* ^c^	0.5 (0.1–1.1)	0.0–3.0
Unspeciated ^b^ *Staphylococcus*	0.5 (0.0–1.7)	0.0–6.4
*Staphylococcus simulans*	0.4 (0.1–0.9)	0.0–2.4
*Streptococcus dysgalactiae*	0.4 (0.1–0.8)	0.0–2.3
*Staphylococcus equorum*	0.4 (0.0–1.4)	0.0–5.4
*Staphylococcus xylosus*	0.4 (0.0–1.1) *	0.0–3.9
*Streptococcus uberis*	0.3 (0.1–0.5)	0.0–1.1
*Corynebacterium xerosis*	0.3 (0.0–1.2)	0.0–4.9
*Corynebacterium bovis*	0.2 (0.0–0.9)	0.0–4.6
*Escherichia coli*	0.2 (0.0–0.7)	0.0–2.3
Unspeciated ^b^ *Aerococcus*	0.2 (0.0–0.7)	0.0–2.2
*Staphylococcus hominis*	0.2 (0.0–0.5)	0.0–1.7
*Lactococcus garvieae*	0.2 (0.0–0.4)	0.0–1.3
*Bacillus licheniformis*	0.1 (0.0–0.5)	0.0–1.8
*Staphylococcus auricularis*	0.1 (0.0–0.3)	0.0–1.3
*Bacillus pumilus*	0.1 (0.0–0.3) *	0.0–1.2
Unspeciated ^b^ *Streptococcus*	0.1 (0.0–0.3) *	0.0–1.0
Unspeciated ^b^ *Micrococcus*	0.1 (0.0–0.2)	0.0–0.8

^a^ Prediction interval represents the range of values that is likely to contain the value of a single new observation, in this case, the microorganism’s prevalence in a new dataset. ^b^ Microorganisms from a given genus that could not be identified to species-level by the MALDI-TOF analysis. ^c^ These microorganisms were reported as *Staphylococcus sciuri*. However, *Staphylococcus sciuri* was recently reassigned to the novel genus *Mammaliicoccus* with *Mammaliicoccus sciuri* as the type species [36]. * Statistically significant difference of the species-specific prevalence between countries (Canada vs. US vs. Brazil), *p* ≤ 0.05. See Appendix A for details.

**Table 4 animals-12-03288-t004:** Prevalence of the different microorganisms isolated from the milk of mammary quarters of dairy cows with clinical mastitis and identified using MALDI-TOF estimated using random meta-analyses conducted on the raw data from 43,924 samples using 8 datasets (Canada, USA and Brazil). Only microorganisms with prevalence ≥0.1% are reported.

Microorganism	Prevalence (95% CI)	95% Prediction Interval ^a^
*Escherichia coli*	11.0 (8.1–14.3)	4.5–19.9
*Streptococcus uberis*	8.5 (5.3–12.2)	1.4–20.4
*Streptococcus dysgalactiae*	7.8 (4.9–11.5)	1.1–19.8
*Staphylococcus aureus*	7.8 (4.4–11.9)	0.4–22.6
*Klebsiella pneumoniae*	5.6 (3.4–8.2)	1.0–13.2
*Staphylococcus chromogenes*	2.3 (0.5–5.3)	0.0–14.0
*Serratia marcescens*	1.0 (0.5–1.7)	0.0–3.4
*Trueperella pyogenes*	0.9 (0.1–2.2)	0.0–6.0
*Corynebacterium bovis*	0.9 (0.1–2.1)	0.0–5.9
*Staphylococcus haemolyticus*	0.7 (0.1–1.8)	0.0–4.2
Unspeciated ^b^ *Staphylococcus*	0.7 (0.0–2.8)	0.0–10.3
*Staphylococcus simulans*	0.6 (0.4–0.8) *	0.1–1.2
*Staphylococcus sciuri* ^c^	0.6 (0.1–1.6)	0.0–4.4
Unspeciated ^b^ *Corynebacterium*	0.6 (0.1–1.4)	0.0–3.8
*Lactococcus garvieae*	0.6 (0.1–1.3)	0.0–2.9
Unspeciated ^b^ *Bacillus*	0.6 (0.0–2.5)	0.0–9.4
*Staphylococcus xylosus*	0.5 (0.0–1.6)	0.0–5.4
*Lactococcus lactis*	0.5 (0.0–1.4)	0.0–4.1
*Streptococcus agalactiae*	0.4 (0.0–2.0)	0.0–7.6
Unspeciated ^b^ *Streptococcus*	0.4 (0.0–1.1)	0.0–3.5
*Enterobacter cloacae*	0.3 (0.2–0.4)	0.2–0.4
*Streptococcus equinus*	0.3 (0.1–0.6)	0.0–1.2
*Staphylococcus epidermidis*	0.3 (0.0–1.2)	0.0–3.8
*Aerococcus viridans*	0.3 (0.0–1.2)	0.0–4.1
*Enterococcus saccharolyticus*	0.3 (0.0–1.0)	0.0–2.4
*Candida rugosa* ^d^	0.3 (0.0–0.9)	0.0–2.6
*Klebsiella oxytoca*	0.3 (0.0–0.8)	0.0–1.8
*Enterococcus faecium*	0.3 (0.0–0.8)	0.0–2.0
*Pasteurella multocida*	0.3 (0.0–0.7) *	0.0–1.5
*Candida krusei* ^d^	0.2 (0.0–0.7) *	0.0–2.1
*Streptococcus gallolyticus*	0.2 (0.0–0.7)	0.0–1.8
*Staphylococcus saprophyticus*	0.2 (0.0–0.6)	0.0–1.7
*Streptococcus lutetiensis*	0.2 (0.0–0.5) *	0.0–1.1
*Bacillus pumilus*	0.2 (0.0–0.5)	0.0–1.3
Unspeciated ^b^ *Prototheca*	0.2 (0.0–0.5)	0.0–1.3
*Staphylococcus hominis*	0.1 (0.0–0.4)	0.0–1.1
*Enterococcus faecalis*	0.1 (0.0–0.3) *	0.0–0.8
*Pseudomonas aeruginosa*	0.1 (0.0–0.3) *	0.0–0.7
*Citrobacter koseri*	0.1 (0.0–0.3)	0.0–1.0
*Corynebacterium amycolatum*	0.1 (0.0–0.3)	0.0–0.9
*Staphylococcus equorum*	0.1 (0.0–0.3)	0.0–1.1
Unspeciated ^b^ *Lactococcus*	0.1 (0.0–0.3)	0.0–1.0
Unspeciated ^b^ *Candida*	0.1 (0.0–0.3)	0.0–0.8

^a^ Prediction interval represents the range of values that is likely to contain the value of a single new observation, in this case, the microorganism’s prevalence in a new dataset. ^b^ Microorganisms from a given genus that could not be identified to species-level by the MALDI-TOF analysis. ^c^ This microorganism was reported as *Staphylococcus sciuri*. However, *Staphylococcus sciuri* was recently reassigned to the novel genus *Mammaliicoccus* with *Mammaliicoccus sciuri* as the type species [36]. ^d^ These microorganisms were reported as *Candida rugosa* and *Candida krusei*. However, *Candida rugosa* was recently reassigned to the novel type species *Diutina rugosa* [37], and *Candida krusei* was reassigned to *Pichia kudriavzevii* [38]. * Statistically significant difference of the species-specific prevalence between countries (Canada vs. US vs. Brazil), *p* ≤ 0.05. See supplementary materials for details.

**Table 5 animals-12-03288-t005:** Microorganisms isolated from dairy cow mammary quarter milk samples and identified using MALDI-TOF, for which relevant literature on associations with somatic cell count (SCC) or clinical mastitis (CM) was available in Medline (PubMed; last search on 16 July 2021).

Microorganism *	No. Articles in Medline	No. Articles Evaluated	No. Relevant Articles	Associated with Increased SCC (Yes/Not Reported)	Associated with CM (Yes/Not Reported)
*Acinetobacter calcoaceticus* ^a^	2	2	1	not reported	yes
*Aerococcus viridans*	1334	15 ^b^	5	yes	yes
*Bacillus licheniformis*	3	3	1	yes	yes
*Bacillus mycoides* ^a^	1	1	1	yes	not reported
*Bacillus pumilus*	5	5	2	yes	not reported
*Bacillus subtilis* ^a^	8	8	1	yes	not reported
*Brachybacterium conglomeratum* ^a^	1	1	1	not reported	yes
*Candida catenulata* ^a,e^	1	1	1	yes	yes
*Candida glabrata* ^a^	4	4	2	not reported	yes
*Candida kefyr* ^a,e^	7	7	3	yes	yes
*Candida krusei* ^e^	15	15	9	yes	yes
*Candida parapsilosis* ^a^	5	5	2	yes	yes
*Candida rugosa* ^e^	6	6	6	yes	yes
*Candida tropicalis* ^a^	10	10	5	yes	yes
*Citrobacter freundii* ^a^	3	3	2	yes	yes
*Corynebacterium stationis* ^a^	1	1	1	yes	not reported
*Enterobacter aerogenes* ^a^	28	28	3	yes	yes
*Enterobacter cloacae*	7	7	4	yes	yes
*Enterococcus avium* ^a^	1	1	1	yes	not reported
*Enterococcus casseliflavus* ^a^	2	2	1	yes	yes
*Enterococcus durans* ^a^	2	2	1	yes	not reported
*Enterococcus gallinarum* ^a^	2	2	1	yes	not reported
*Enterococcus hirae* ^a^	5	5	3	yes	yes
*Geotrichum capitatum* ^a^	2	2	1	yes	yes
*Histophilus somni* ^a^	17	17	3	yes	yes
*Klebsiella oxytoca*	14	14	7	yes	yes
*Klebsiella pneumoniae*	124	3 ^c^	2	yes	yes
*Klebsiella variicola* ^a^	2	2	1	not reported	yes
*Lactococcus garvieae*	11	11	4	yes	yes
*Lactococcus lactis*	38	30 ^b^	6	yes	yes
*Listeria innocua* ^a^	1	1	1	not reported	yes
*Listeria monocytogenes* ^a^	59	51 ^b^	7	yes	yes
*Mannheimia granulomatis* ^a^	21	21	1	yes	not reported
*Nocardia farcinica* ^a^	6	6	2	yes	yes
*Nocardia puris* ^a^	1	1	1	yes	yes
*Pasteurella multocida*	29	29	4	yes	yes
*Proteus mirabilis* ^a^	2	2	1	not reported	yes
*Proteus vulgaris* ^a^	2	2	2	yes	yes
*Pseudomonas fluorescens* ^a^	3	3	1	yes	not reported
*Raoultella ornithinolytica* ^a^	1	1	1	not reported	yes
*Raoultella planticola* ^a^	3	3	1	not reported	yes
*Staphylococcus agnetis* ^a^	12	12	8	yes	yes
*Staphylococcus arlettae* ^a^	5	5	3	yes	yes
*Staphylococcus auricularis*	2	2	2	yes	yes
*Staphylococcus capitis* ^a^	12	12	8	yes	yes
*Staphylococcus devriesei* ^a^	8	8	5	yes	yes
*Staphylococcus equorum*	21	21	8	yes	yes
*Staphylococcus fleurettii* ^a,d^	9	9	2	not reported	yes
*Staphylococcus gallinarum* ^a^	4	4	1	yes	yes
*Staphylococcus haemolyticus*	88	49 ^b^	3	yes	yes
*Staphylococcus hominis*	33	13 ^b^	4	yes	yes
*Staphylococcus intermedius* ^a^	23	23	8	yes	yes
*Staphylococcus lugdunensis* ^a^	2	2	1	not reported	yes
*Staphylococcus microti* ^a^	3	3	2	yes	yes
*Staphylococcus succinus* ^a^	7	7	4	yes	yes
*Stenotrophomonas maltophilia* ^a^	3	3	1	yes	yes
*Streptococcus canis* ^a^	7	7	3	yes	not reported
*Streptococcus equinus*	112	5 ^b^	1	not reported	yes
*Streptococcus lutetiensis*	1	1	1	not reported	yes
*Streptococcus oralis* ^a^	1	1	1	yes	not reported
*Streptococcus salivarius* ^a^	7	7	1	yes	not reported
*Weissella cibaria* ^a^	4	4	1	yes	yes
*Weissella paramesenteroides* ^a^	2	2	1	not reported	yes
*Yersinia pseudotuberculosis* ^a^	4	4	2	yes	yes

^a^ Microorganisms with prevalence <0.1% (either in bovine milk samples of apparently healthyhealthy lactating quarters and/or from quarters with CM). ^b^ Restricting evaluation of articles to articles with the microorganism mentioned in the title or abstract. ^c^ Restricting evaluation of articles to review articles. ^d^ These microorganisms were reported as *Staphylococcus sciuri* or *Staphylococcus fleurettii*. However, *Staphylococcus sciuri* was recently reassigned to the novel genus *Mammaliicoccus* with *Mammaliicoccus sciuri* as the type species, and *Staphylococcus fleurettii* was reassigned to *Mammaliicoccus fleurettii* [36]. ^e^ These microorganisms were reported as *Candida catenulata, Candida kefyr, Candida krusei and Candida rugosa*. However, *Candida catenulata* was recently reassigned to the novel type species *Diutina catenulate* [37], *Candida kefyr* was reassigned to *Kluyveromyces marxianus* [39], *Candida krusei* was reassigned to *Pichia kudriavzevii* [38], *Candida rugosa* was reassigned to *Diutina rugosa* [37]. * The following 21 bacterial species, already described as important mastitis-causing pathogens in Current Concepts of Bovine Mastitis [2], were not included in the literature search: *Staphylococcus aureus*, *Streptococcus agalactiae*, *Streptococcus uberis*, *Streptococcus dysgalactiae*, *Mycoplasma bovis*, *Corynebacterium bovis*, *Escherichia coli*, *Enterococcus faecalis*, *Enterococcus faecium*, *Staphylococcus chromogenes*, *Staphylococcus hyicus*, *Staphylococcus warneri*, *Staphylococcus epidermidis*, *Staphylococcus cohnii*, *Staphylococcus simulans*, *Staphylococcus xylosus*, *Staphylococcus sciuri*, *Staphylococcus saprophyticus*, *Pseudomonas aeruginosa*, *Trueperella pyogenes*, and *Bacillus cereus*.

## Data Availability

The data and statistical analyses are available at https://doi.org/10.5683/SP3/VOUUYO.

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
