# Peer review of "Diagnosing Intramammary Infection: Meta-Analysis and Mapping Review on Frequency and Udder Health Relevance of Microorganism Species Isolated from Bovine Milk Samples"

_animals, 2022, doi:10.3390/ani12233288_

Round 1

Reviewer 1 Report

A very interesting and useful article. This meta-analyis confirm utility of MALDI-TOF methods identification of bacterial/fungal microorganism associated with mastitis. Moreover, collects data about intramammary infections. I agree with Authors, that little literature regarding their udder health  relevance is available. Thus, meta-analyzis data is very important to understanding of natural beeing of non-pathogenic bacteria. Moreover, data is assiociated with somatic cel count. This study provides guidance not only to veterinarians but also to veterinary diagnostic laboratory and  scientists studying mastitis. A total of 294 different species of microorganisms recovered from milk samples were identified, in tables and in Supplementary Table 1.

Minor Comments

Lack of information about Mycoplasma sp.

Line 26: Authors write that MALDI-TOF „can identify many different microorganisms cultured from bovine milk samples to species-level. However, all bacteria from Table 3, Table 4, 5 are successfully identified by VITEK2 (cards GP, GN, CBC, BCL, and YST). It should be mentioned in text and conclusion.

Authors using 93,353 milk samples from different countries, but mainly from US and Canada. It is recognized that the capacity to perform these experiments many not be available, however.

Author Response

Response to Reviewer 1 Comments

Point 1 : Lack of information about Mycoplasma sp.

Response 1 : We appreciate a lot your feedback on our manuscript. Indeed, we have not much information on the Mycoplasma spp., as well as for other genera. The performance of the standard bacteriological culture using blood agar are limited, when comes to identification of the species of Mycoplasma spp. The fact that the selective media and/or the appropriate growth conditions for this microorganism were not used in any study, included in our analyses, gave us no species-level identification of the species of Mycoplasma genus. Regarding not obtaining Mycoplasma organisms in the datasets shared with us, we mentioned on lines 430-433 that it was due to the growth conditions used. Regarding reviewing the literature on udder health impact of these organisms, we mentioned on line 211-216 that they are already well-recognized udder health pathogens.

Point 2 : Line 26: Authors write that MALDI-TOF „can identify many different microorganisms cultured from bovine milk samples to species-level. However, all bacteria from Table 3, Table 4, 5 are successfully identified by VITEK2 (cards GP, GN, CBC, BCL, and YST). It should be mentioned in text and conclusion.

Response 2 : We agree with your comment, many different methods can be used for bacterial identification. We completed the sentence on line 89-92 to highlight this. The objective of this sentence, however, was not to provide a comprehensive list of all the available methods (which would be nearly impossible). Your comment on Vitek2 made us realized, though, that we did not mentioned the type of MALDI-TOF instruments used for generating the MALDI-TOF results in the datasets that were shared with us. Indeed, both the Vitek and Biotyper systems (or other systems) were considered for data obtention. However, only datasets generated using the Bruker Daltonics Inc MALDI-TOF equipment were submitted. We added a precision in lines 275-276 (Results section) on this matter.

Point 3 : Authors using 93,353 milk samples from different countries, but mainly from US and Canada. It is recognized that the capacity to perform these experiments many not be available, however.

Response 3 : We support your opinion, and understand that our results could not be applied to all regions/countries worldwide. Please, find the acknowledgement on this matter in lines 502 to 506 (Discussion section). As mentioned on lines 146-150, we tried to obtained results from all over the world to have more representative picture. We did received responses from scientists from Europe, Australia, and New-Zealand, but the datasets that could have been shared by these scientists did not met our inclusion criteria. Some were cow-level data, and most were from sampling where only high SCC and/or cows with mastitis history were selected. Thus, these datasets would not allow to compute unbiased measures of prevalence.

Reviewer 2 Report

The study by Kurban and the coauthors deals with the determination of diversity and proportion of different microbes in healthy cows and those with mastitis using MALDI-TOF data. 

The study design, data collection section and the approach to deduce the results/ conclusion are impressive.

What i see missing in the manuscript is information regarding methods on mastitis detection used around the globe. For a reader not familiar with the topic, i recommend to adding a section on methodologies in the Introduction. The following reference (but not limited to) can be taken into account:

  1. Barnum, D.A.; Newbould, F.H. The Use of the California Mastitis Test for the Detection of Bovine Mastitis. Can. Vet. J. Rev. Vet. Can. 1961, 2, 83–90.
  2. Ruegg, P.L. A 100-Year Review: Mastitis detection, management, and prevention. J. Dairy Sci. 2017, 100, 10381–10397.
  3. Norberg, E. Electrical conductivity of milk as a phenotypic and genetic indicator of bovine mastitis: A review. Livest. Prod. Sci. 2005, 96, 129–139.
  4. Sheldrake, R.; McGregor, G.; Hoare, R. Somatic Cell Count, Electrical Conductivity, and Serum Albumin Concentration for Detecting Bovine Mastitis. J. Dairy Sci. 1983, 66, 548–555.
  5. Kasai, S.; Prasad, A.; Kumagai, R.; Takanohashi, K. Scanning Electrochemical Microscopy-Somatic Cell Count as a Method for Diagnosis of Bovine Mastitis. Biology 2022, 11, 549. https://doi.org/10.3390/biology11040549

Author Response

Response to Reviewer 2 Comments

Point 1 : The study by Kurban and the coauthors deals with the determination of diversity and proportion of different microbes in healthy cows and those with mastitis using MALDI-TOF data. 

The study design, data collection section and the approach to deduce the results/ conclusion are impressive.

Response 1 : Thank you for your positive comments. We tried to do our best to provide interesting results

Point 2 : What i see missing in the manuscript is information regarding methods on mastitis detection used around the globe. For a reader not familiar with the topic, i recommend to adding a section on methodologies in the Introduction. The following reference (but not limited to) can be taken into account:

  1. Barnum, D.A.; Newbould, F.H. The Use of the California Mastitis Test for the Detection of Bovine Mastitis. Can. Vet. J. Rev. Vet. Can. 1961, 2, 83–90.
  2. Ruegg, P.L. A 100-Year Review: Mastitis detection, management, and prevention. J. Dairy Sci. 2017, 100, 10381–10397.
  3. Norberg, E. Electrical conductivity of milk as a phenotypic and genetic indicator of bovine mastitis: A review. Livest. Prod. Sci. 2005, 96, 129–139.
  4. Sheldrake, R.; McGregor, G.; Hoare, R. Somatic Cell Count, Electrical Conductivity, and Serum Albumin Concentration for Detecting Bovine Mastitis. J. Dairy Sci. 1983, 66, 548–555.
  5. Kasai, S.; Prasad, A.; Kumagai, R.; Takanohashi, K. Scanning Electrochemical Microscopy-Somatic Cell Count as a Method for Diagnosis of Bovine Mastitis. Biology 2022, 11, 549. https://doi.org/10.3390/biology11040549

Response 2 : Indeed, this is an important point. The information on different mastitis diagnostic methods were added in the manuscript. Please see the lines from 70 to 92. All suggested references were used, as well as some other studies which explain one or another method.
